# Novel Competitive ELISA Utilizing Trimeric Spike Protein of SARS-CoV-2, Could Identify More Than RBD-RBM Specific Neutralizing Antibodies in Hybrid Sera

**DOI:** 10.3390/vaccines12080914

**Published:** 2024-08-13

**Authors:** Petros Eliadis, Annie Mais, Alexandros Papazisis, Eleni K. Loxa, Alexios Dimitriadis, Ioannis Sarrigeorgiou, Marija Backovic, Maria Agallou, Marios Zouridakis, Evdokia Karagouni, Konstantinos Lazaridis, Avgi Mamalaki, Peggy Lymberi

**Affiliations:** 1Immunology Laboratory, Immunology Department, Hellenic Pasteur Institute, 11521 Athens, Greece; a.papazisis@pasteur.gr (A.P.); e.loxa@pasteur.gr (E.K.L.); sarijon@pasteur.gr (I.S.); klazaridis@pasteur.gr (K.L.); 2Biotechnology Unit, Hellenic Pasteur Institute, 11521 Athens, Greece; aldimitriadis@pasteur.gr (A.D.); amamalaki@pasteur.gr (A.M.); 3Laboratory of Molecular Biology and Immunobiotechnology, Immunology Department, Hellenic Pasteur Institute, 11521 Athens, Greece; maisannie@hotmail.com; 4Institut Pasteur, Unité de Virologie Structurale, Université Paris Cité, CNRS-UMR3569, 75724 Paris, France; marija.backovic@pasteur.fr; 5Immunology of Infection Laboratory, Microbiology Department, Hellenic Pasteur Institute, 11521 Athens, Greece; mariaagallou@pasteur.gr (M.A.); ekaragouni@pasteur.gr (E.K.); 6Structural Neurobiology Research Group, Laboratory of Molecular Neurobiology and Immunology, Department of Neurobiology, Hellenic Pasteur Institute, 11521 Athens, Greece; mzouridakis@pasteur.gr

**Keywords:** SARS-CoV-2, surrogate viral neutralization assay (sVNA), trimeric S(6P)-HexaPro, non-RBD-RBM neutralizing antibodies (nAbs)

## Abstract

Since the initiation of the COVID-19 pandemic, there has been a need for the development of diagnostic methods to determine the factors implicated in mounting an immune response against the virus. The most promising indicator has been suggested to be neutralizing antibodies (nAbs), which mainly block the interaction between the Spike protein (S) of SARS-CoV-2 and the host entry receptor ACE2. In this study, we aimed to develop and optimize conditions of a competitive ELISA to measure serum neutralizing titer, using a recombinant trimeric Spike protein modified to have six additional proline residues (S(6P)-HexaPro) and h-ACE2. The results of our surrogate Virus Neutralizing Assay (sVNA) were compared against the commercial sVNT (cPass, Nanjing GenScript Biotech Co., Nanjing City, China), using serially diluted sera from vaccinees, and a high correlation of ID_50–90_ titer values was observed between the two assays. Interestingly, when we tested and compared the neutralizing activity of sera from eleven fully vaccinated individuals who subsequently contracted COVID-19 (hybrid sera), we recorded a moderate correlation between the two assays, while higher sera neutralizing titers were measured with sVNA. Our data indicated that the sVNA, as a more biologically relevant model assay that paired the trimeric S(6P) with ACE2, instead of the isolated RBD-ACE2 pairing cPass test, could identify nAbs other than the RBD-RBM specific ones.

## 1. Introduction

The great interest of the scientific community in COVID-19 has resulted in the understanding, to a large extent, of the viral infection mechanisms. The trimeric Spike (S) protein of SARS-CoV-2 is responsible for human Angiotensin Converting Enzyme 2 (hACE2) receptor binding and the consequent fusion of the virus with the host cell membranes [1]. The S monomer consists of the S1 and S2 subunits. The S1 subunit contains the receptor-binding domain (RBD), the N-terminal domain (NTD) and the intermediate region spike subdomain 1 and 2 (SD1 and SD2). The Receptor Binding Motif (RBM) is located within the RBD fragment, which contains the residues that enable ACE2 binding. The S2 subunit has been implicated in cell entrance [2,3]. 

Over the progress of the pandemic, and with the commercialization of several vaccines, the detection of neutralizing antibodies (nAbs) as a potential marker of effective immunity has become increasingly important for the estimation of vaccine trial success and the evaluation of host immunity against infection. It is believed that the majority of nAbs target the RBD fragment, mostly through recognition of the RBD-RBM [4,5]. The S1–specific nAbs that do not bind the RBD-RBM are the next potent nAbs identified, followed by nAbs able to bind to S protein only in its trimeric conformation, and finally nAbs against the S2 subunit [4,6,7]. Notably, recent studies have reported the existence of nAbs in sera from both natural infection and vaccination, specific for epitopes outside the RBD-RBM region that were implicated in the RBD-ACE2 interaction. In particular, some publications described SD1– and NTD-targeting Abs that were able to change the configuration of S trimer [5], thereby neutralizing or enhancing ACE2 binding by altering the “up-to-down” or “down-to-up” RBD conformation [4,8,9,10]. Furthermore, it is reported that an anti-S2 specific Ab can induce the RBD in the up conformation, thus enhancing ACE2 binding [10]. These conformational changes of S protein could also be affecting the specificity or the affinity strength of nAbs that compete with ACE2. Moreover, antibodies bound to both conformation states of RBD (“up” and “down”) have now been reported [11,12]. Another study referred to a nAb that acted by locking the S-trimer in a closed conformation, by bridging two adjacent RBDs on a single S-trimer [13]. A study by Meng and colleagues reported nAbs that bind RBD non-RBM area, acting through a neutralizing mechanism that destructs the S trimer [14]. Finally, there has been a report about a neutralizing antibody in a convalescent serum targeting the NTD that disrupts the S-trimer by altering the “up-to-down” RBD conformation, while these findings were also supported by reactivity studies with the recombinant S(6P) in ELISA [15]. 

Multiple methods have been developed for the quantification of nAbs requiring a live virus and cells expressing hACE2 (virus neutralizing tests, VNTs), the gold standard of which is the Plaque Reduction Neutralization Test (PRNT) [6,16,17]. This technique evaluates the total neutralizing ability of the tested sera samples, including nAbs and other inhibitory molecules that contribute to viral infection. In addition, multiple pseudovirus assays (pVNTs) have been developed, the results of which are highly correlated with those of live VNTs [6,17]. Although these tests are of utmost significance, they require highly trained personnel, increased biocontainment laboratories, and long incubation periods, thereby rendering them unsuitable for evaluating large amounts of samples for diagnostic purposes. Thus, efforts have been made to develop enzyme immunoassays as an alternative to measuring neutralizing titers. Until now, several “in-house” competitive ELISAs that determine the effect of nAbs on the interaction of RBD and ACE2, have been experimentally developed and termed surrogate virus neutralizing tests (sVNT), while there are others commercially available. For the application of these ELISA-based methods, recombinant tagged proteins of the antigens RBD, S1 and prefusion-stabilized trimeric S were used [6,17,18,19,20,21], with the majority of them joining together RBD with the ACE2 receptor, while some used the S1–ACE2 pair [18,22,23,24,25,26,27,28,29]. To the best of our knowledge, there are only four commercially available sVNT kits [30,31,32,33], but none using trimeric HexaPro Spike molecule (S(6P)) [19]. Abe and colleagues compared the ability of RBD or S-trimer (stabilized by two prolines) antigens to interact with ACE2 and nAbs and concluded that both worked well [18]. Another study stated that similar experiments showed that when S-trimer was used as the antigen to interact with ACE2 and nAbs, the inhibition was lower than that observed with RBDs [28]. Notably, a recombinant Spike native-like trimeric protein was successfully used together with ACE2 in an inhibition assay, to evaluate the ability of this molecule to be recognized by nAbs from sera of COVID-19 convalescents and vaccinated animals [34]. 

The idea to use a native-like trimeric Spike molecule in an inhibition assay is appealing. Such a choice seems to cover the mode of action of all protein molecules (including nAbs) or other factors that target not only RBD fragments, but also S trimer and S1 and S2 domains of SARS-CoV-2 S protein, and thus contribute directly or indirectly to the inhibition of trimeric S protein binding to ACE2. A competitive ELISA that pairs S trimer and ACE2 recombinant molecules seems to offer advantages as a biologically more relevant model for the measurement of nAbs activity than the RBD-ACE2 interaction-based assay.

In the present study, we aimed to develop a competitive ELISA for the quantification of nAbs, generated against the S protein and directly or indirectly implicated with the inhibition of the RBD binding to ACE2. To this end, we chose to use the soluble recombinant trimeric Spike protein HexaPro, a prefusion stabilized spike variant with six additional proline residues, which is strep tagged (S(6P)-StrepTag). Structural experiments well characterized and confirmed that S(6P), as all other class I fusion proteins, retains the trimeric prefusion spike conformation [19] and a glycosylation motif similar to the native spike antigen [35], while it was reported to have less RBDs in an open conformation than S(2P) [36]. Hsieh and colleagues confirmed that this molecule manifests higher expression levels and greater stability than the S(2P) construct, by being resistant to heat stress [19]. Despite the increased stability of trimeric S(6P), its RBDs continue to occupy multiple positions [19,37,38]. Interestingly, data from several studies have confirmed similar functional characteristics in the interaction of ACE2 with HexaPro [38,39], to those of the interaction of the SARS-CoV-2 spike ectodomain as part of the viral membrane [40,41]. So far, HexaPro has been used as an immobilized antigen in several serological studies, showing similar or increased affinity compared to S(2P) and spike subunits for polyclonal antibodies from convalescent or vaccinated individuals [19,34,35,42,43]. We have opted to use this molecule for the first time, to our knowledge, in an inhibition ELISA optimized for the detection of SARS-CoV-2 nAbs. As a source of nAbs, we used sera from subjects vaccinated with the mRNA vaccine BNT162b2 (BioNTech, Marburg, Germany/Pfizer, New York, NY, USA) [44] and from fully vaccinated individuals (3 doses) who contracted COVID-19 after having finished their vaccination program (hybrid sera). We applied our in-house competitive ELISA to investigate the status of nAbs titers in vaccine recipients and in convalescent patients vaccinated with three doses of Pfizer vaccine (vaccinated patients with COVID-19) and compared the results with those of an RBD-sVNT (cPass, Genscript). Our findings demonstrated the suitability and underlined the necessity of using a spike trimer molecule to develop a surrogate VNT, capable of measuring the activity of all nAbs that directly or indirectly block the RBD-ACE2 interaction. 

## 2. Material and Methods 

### 2.1. Sera Samples

A total of 181 sera samples from the Health Care Workers in Hellenic Pasteur Institute, obtained at various time points before (47 samples) and after vaccination (120 samples from 29 subjects + 14 hybrid samples) with Pfizer’s SARS-CoV-2 vaccine (BNT162b2, BioN-Tech/Pfizer), were included in this study. The 29 vaccinated were healthy individuals of Greek origin aged 23–63 (21/29 female and 8/29 male). Fourteen of the subjects were classified as young adults (23–40) while 15 were middle-aged (41–63). Nine of the 14 patients with COVID-19 were female, five of them were male. All vaccinees infected with SARS-CoV-2 became mildly or moderately ill, with no one being hospitalized. In addition, pre-COVID-19 sera (*n* = 76) from DIAGNOSTIKI ATHINON (Clinical and Research Laboratory, Athens, Greece), stored at −80 °C, were used as negative controls and for the determination of the cut off of the ELISA assays. 

The 29 vaccine recipients were vaccinated with 3 doses of Pfizer vaccine from January to December 2021. Subjects’ (s) blood samples (S) were collected at several time points, characterized as S1–S10: the first, before the first vaccination (S1), and subsequently 21 days after the first dose (S2); the day of second dose, 21 days (S3), 60–65 days (S4), 190–200 days (S5), and 240–260 days after the second dose (S6); the day of third dose, and 21 days (S7), 60–65 days (S8), 90–120 days (S9), and ~200 days (S10) after third dose. For fourteen vaccinated individuals who contracted COVID-19, at least 200 days elapsed after the third vaccine dose; blood samples (hybrid sera) were collected 15–25 days (S11) after PCR confirmation (Table 1). All sera samples were labeled and archived using a coding system; a unique code was attributed to each participant (s/S). The sera, prepared from whole blood, were divided into aliquots, stored at −80 °C and thawed only once for utilization in ELISA assays. 

The use of samples in this study was in accordance with the Declaration of Helsinki and International Conference for Harmonization for Good Clinical Practice. All subjects gave their written informed consent, before participating in the study. Subjects’ data were kept confidential in accordance with the rules of the General Data Protection Regulation. All inclusion criteria for the sera donors are described in the Appendix A. 

### 2.2. Recombinant Protein Antigens

The pcDNA3–sACE2(WT)–8his (amino acids 1—615) was a gift from Erik Procko (Addgene plasmid #149268; http://n2t.net/addgene:149268, accessed on 19 March; RRID:Addgene_149268).

The pcDNA3.1 plasmid encoding the SARS-CoV-2 S ectodomain (amino acids 1 to 1208) with the six proline mutations (6P), F817P, A892P, A899P, A942P, K986P, and V987P, were introduced to stabilize and increase protein yields, followed by a fold-on trimerization motif and tags (8 x HisTag, StrepTag, and AviTag) produced by L. Grzelak et al. Prof. Felix Rey (Structural Virology Research Unit, Institute Pasteur Paris). After S(6P) was produced from pcDNA3.1-S(6P) and purified, its trimeric formation was further evaluated by the investigators [45,46]. We used the above plasmid construct to produce recombinant S(6P) protein. 

The ACE2 glycoprotein and the trimeric S(6P) glycoprotein were produced by transient transfection of exponentially growing Expi293 suspension cells (Thermo Fisher Scientific, Waltham, MA, USA) using FectoPRO transfection reagent (Polyplus, Illkirch, France) following the manufacturer’s guidelines. Recombinant ACE2 protein was purified by affinity chromatography using the Ni Sepharose excel resin (ProtinoNi-NTA column, Machrey-Nagel, Düren, Germany), whereas the recombinant SARS-CoV-2 protein (S-6P) was purified by affinity chromatography using Strep-Tactin XT 4flow column (IBA Lifesciences, Göttingen, Germany) according to the manufacturer’s instructions. The eluates were analyzed in 4–20% gradient precast protein gels (Nippon Genetics, Düren, Germany), pooled appropriately and buffer exchanged with PBS, pH 7.2.

S(6P) trimer formation was detected in the final product eluent of the anti-strep column purification. The trimeric form of the eluted S(6P) protein was confirmed (Appendix A) by gel filtration analysis (Appendix A).

Both antigens purity and integrity was finally determined by SDS–PAGE and the protein concentration was measured by Bradford (Biorad, Hercules, California, CA, USA), using BSA (Bovine Serum Albumin) for standard curve, and by the Protein Concentration Calculator webserver (https://www.aatbio.com/tools/calculate-protein-concentration, accessed on 19 March 2021) using the absorbance value at 280 nm, measured in an UV/Vis spectrophotometer (Smart Spec^TM^ Plus, BIO RAD, Hercules, CA, USA), the extinction coefficient, and the molecular weight (MW) of the protein obtained from the Expasy webserver (https://web.expasy.org/protparam/, accessed on 19 March 2021). Values obtained by Bradford were slightly higher than those obtained by the calculator, with the latter finally chosen in calculations. Recombinant proteins were divided into aliquots and stored at −80 °C so that they could be thawed only once (Appendix A). The purified S(6P)-StrepTag product was already used for serological studies [43].

### 2.3. Surrogate Virus Neutralization Assay (sVNA) Development

#### 2.3.1. Direct ELISA Analysis

Optimization of the inhibition assay (sVNA) proceeded first by selecting the appropriate blocking and dilution buffer. The impact of no-fat milk (n.f.-milk) and Fetal Bovine Serum (FBS, -IgGs) was studied in S(6P) and hACE2 interaction as well as between specific human (h) IgG antibodies and S(6P), in direct ELISA. Briefly, hACE2 protein and S(6P), diluted in PBS were used to coat 96–well half area high-binding plates. Both n.f.-milk and FBS were utilized for the blocking of non-specific binding sites and in dilution solutions. The binding of S(6P) to immobilized ACE2 was determined with anti-strep-tag (MAB-Immo) and anti-mouse/HRP conjugate while the binding of h-IgGs to immobilized S(6P) with anti-hIgG(Fcγ)-HRP, followed by the detection of peroxidase activity using TMB substrate reagent and stop solution. Further details are described in Appendix A. 

#### 2.3.2. Inhibition ELISA Analysis

Human ACE2 protein diluted in PBS was coated in half area 96–well high binding plates (#3690, Corning, New York, NY, USA). Non-specific binding sites were blocked with blocking solution. Trimeric S(6P)-StrepTag antigen (Ag), anti-StrepTag (mouse mAb-Immo) and anti-mouse/HRP conjugate were mixed and pre-incubated with the tested sample of serum for 1 h at 37 °C. The pre-incubated mixture was then incubated in the coated ELISA plate for 1 h at R.T., where the free S(6P) Ag, as well as the Ag’s molecules that bound to non-neutralizing antibodies, interacted with hACE2 and were captured on the plate (Figure 1). Incubation was followed by washing steps to remove unbound serum components, including S(6P)-StrepTag complexed with mAb-Immo and conjugate/HRP. The chromogenic reaction was quantified after addition of TMB substrate and stop solution. The absorbance of the samples was measured at 450 nm. Further details are described in the Appendix A.

### 2.4. cPass SARS-CoV-2 Neutralization Antibody Detection Kit (Genscript)

Sera samples were tested for neutralizing activity with the sVNT (surrogate Viral Neutralizing Test) cPass Assay (Genscript) at a final dilution of 1:20 or at consecutive dilutions for the titration of the neutralizing activity, according to the manufacturer’s instructions. Further details are described in Appendix A.

### 2.5. Statistical Analysis

All statistical analyses were performed using GraphPad Prism software version 6.01 (GraphPad Software, Boston, MA, USA). Microsoft Excel was used to generate some graphs (Appendix A).

## 3. Results

### 3.1. Development of the Surrogate Virus Neutralization Assay (sVNA)

In general, the principle of surrogate ELISA based VNTs includes the following: (a) a pre-incubation step for the immobilized antigen (RBD, S1 or spike trimer) with the serum sample to be tested and then addition of soluble ACE2-tagged protein, which is bound by the free antigen molecules and detected enzymatically; (b) a pre-incubation step for the solubilized tagged antigen with the serum sample under test and then addition to the ACE2 coated ELISA plate, whereas the free antigen molecules are bound and detected enzymatically. Both give a low optical signal when nAbs are present. In the sVNA immunoassay we developed, we opted for the second protocol (Figure 1), using the (S6P)-StrepTag trimer as antigen.

#### 3.1.1. Optimizing the Surrogate Virus Neutralization Assay (sVNA)

Recombinant SARS-CoV-2 trimeric spikes with six additional proline mutations (S(6P), HexaPro), with tags at the C-terminus (Section 2.2) including StrepTag (WSHPQFEK), were used for this work [45,46]. Based on our preliminary experiments, the optimization of the neutralization assay first focused on the selection of the appropriate blocking and dilution agent (between FBS and n.f.-milk) (Appendix A). Our results showed that the optimal concentration of immobilized ACE2 was 2.75 μg/mL in a 50 μL volume/well (Appendix A), while FBS was the appropriate agent in the blocking and dilution solutions instead of n.f.-milk, for our experimental setup (Appendix A).

Subsequently, we used the S(6P)-StrepTag antigen at an optimal concentration combined with both agents to compare the final inhibitory effect produced with sVNA-S(6P) when testing titrated sera from vaccinated individuals (Appendix A). Our results have shown that utilization of S(6P) in combination with FBS performed better, producing a more desirable inhibition curve than n.f.-milk (Appendix A).

Following the serial dilution protocol for the S(6P)/FBS-sVNA (Material and Methods, Section 2.3.2), we also tested different concentrations from the optimal one (115 ng/mL) of S(6P)-StrepTag in the pre-incubation step with sera from vaccinated individuals, which resulted in the shift of the initial inhibition curve (Figure 2). More specifically, keeping the S(6P) concentration lower than 115 ng/mL achieved higher sensitivity (80 ng/mL, grey line), while using higher S(6P) concentration (150 ng/mL, blue line) resulted in lower sensitivity of the assay. In parallel, we decided to test our optimization steps with the cPass, a widely used commercial inhibition kit which we also used for the validation of our assay. The sVNT of Genscript (cPass, approved by FDA and CE) detects nAbs that block the interaction between the isolated RBD of the SARS-CoV-2 spike glycoprotein and the ACE2 [22]. An inhibition curve was produced (Section 2.4) and compared with the sVNA-generated inhibition curves. The experiment was repeated with three different sera. The data showed an almost equivalent inhibition curve between 115 ng/mL S(6P)-sVNA and cPass with ID_50_ values of 696.6 and 722.3 respectively (Figure 2), although after the experiments were completed, the ID_50_ values of cPass tended to be slightly higher for the majority of vaccinees’ sera.

#### 3.1.2. Testing Sera Samples with sVNA

Following the optimized protocol (Section 2.3.2), a dose-dependent inhibition occurred when sera from vaccinated individuals and hybrid sera were tested with sVNA at serial dilutions. Figure 3A–C shows the percent inhibition of sera obtained from five vaccinated individuals 21 days after the administration of the 1st, 2nd, and 3rd dose of the vaccine (sera samples S2, S3, and S7). The initial 1:20 dilution showed an extremely high percentage of inhibition (>90%) for the S3 samples, while the S2 samples gave moderate results. Interestingly, all S7 sera samples exhibited >90% inhibition at higher titers than S3 samples, as recorded by the higher titer ID_50_ values (Figure 3D). Similar inhibition results are demonstrated in Appendix A (inhibitory curves of all sera samples throughout the vaccination period of a person that was infected with SARS-CoV-2. The above results are consistent with those of other studies [47,48]. 

### 3.2. Validation and Characteristics of the sVNA 

Sera obtained from vaccinated individuals, hybrid sera and negative controls (pre-COVID-19 and pre-vaccinated sera), were tested with sVNA for their antibodies’ ability to neutralize the trimeric S(6P)-StrepTag-ACE2 interaction, and the results produced were correlated with those of the RBD-sVNT cPass inhibition assay. A diagram of how we performed the following experiments is shown in Figure 4.

#### 3.2.1. Robustness of Negative Sera Samples Evaluation—Reproducibility of sVNA 

Pre-COVID-19 sera (*n*:76, true negative sera, obtained in 2018) were tested with sVNA at a final serum dilution of 1:20, in two independent experiments (exp.1 and exp.2) and pre-vaccination sera (*n*:47, presumed negative sera, obtained on the 19–21 February 2021) were tested once. The results from both groups of sera showed that they were concentrated at, or below, 20% inhibition (Figure 5A). The mean values ± SD were 2.04% ± 5.6% (exp.1), 2.41% ± 5.8% (exp.2) and 2.8% ± 5.8%, respectively. These results suggested that a cut-off value of 20% would allow discrimination between negative and positive sera when tested with the sVNA.

To assess the reproducibility of sVNA in both sera of pre-vaccinated (*n* = 13) and vaccinated individuals (*n* = 27), their inhibitory activity was measured on different dates at a dilution of 1:20, giving a strong correlation (R^2^ = 0.9508) (Figure 5B). Also, regression-correlation analysis of ID_50_ values of % inhibition for twenty-two inoculated sera tested in two different experiments gave a high correlation of determination (R^2^ = 0.9343) (Figure 5C). 

#### 3.2.2. Correlation between sVNA and the cPass (Genscript)

There are multiple comparative and correlation studies between different methods for the detection and quantification of human neutralizing SARS-CoV-2 Abs and cPass, including virus neutralization tests (VNT), the PRNT (Plaque Reduction Neutralization Test), and the FRNT (Focus Reduction Neutralization Test) [22,49,50,51,52,53,54], as well as the viral pseudotype test (pVNT) [49,55,56]. In general, these studies reported very good agreement between IC_50_–IC_90_ titers generated by VNTs/pVNTs and cPass inhibition values in terms of specificity and sensitivity (when 1:20 serum dilution was used).

We used the commercial cPass kit to validate our “in-house” ELISA inhibition assay (sVNA) by comparing (a) the ID_90_, ID_50_, and ID_30_ inhibitory titers and (b) the inhibition values at 1:20 serum dilution produced by the two assays. Correlation analysis between the values of the two inhibition assays was carried out using linear regression. Thirty-two sera from vaccinated individuals were tested with the two assays at final dilution 1:20 (Figure 6A), giving a strong coefficient of determination (R^2^ = 0.9129, *p* < 0.0001).

In addition, twenty-three sera from vaccinated individuals were serially diluted with both sVNA and cPass and the ID_90_, ID_50_, and ID_30_ dilution inhibition titer values were determined and compared. Linear regression analysis (Figure 6B–D) gave a high coefficient of determination for ID_90_ and ID_50_ (of R^2^ = 0.9614 and R^2^ = 0.9195 respectively, *p* < 0.0001) and slightly weaker one for ID_30_ (R^2^ = 0.8828, *p* < 0.0001). These data, resulting from testing vaccinees’ sera, showed that sVNA gives comparable results to cPass. Additionally, a head-to-head comparison of the inhibitory ID_50_ values of the twenty three vaccinees’ sera (Figure 6C), showed that for most of them a higher sensitivity was achieved with cPass. In more detail, the sVNA and cPass titration curves of eight of these sera and the ID_50_ inhibitory values that go with them are shown in Appendix A.

Finally, we tested the inhibitory effect of the WHO international standard for anti-SARS-CoV-2 immunoglobulin (NIBSC (21/234) with the sVNA and cPass assays. The resulting ID_50_ values were similar (267.5 and 293.8 respectively), with a ratio of cPass-ID_50_: sVNA-ID_50_ = 1.1 (Appendix A).

### 3.3. Evaluation of nAbs Titers in Vaccinees and Hybrid Sera with sVNA

Sera (*n* = 120) with different levels of nAbs because of varied chronological blood sampling points (S2–S9), as well as 14 hybrid sera (S11), were tested with sVNA at a final dilution of 1:20 for their ability to inhibit the S(6P)-ACE2 interaction (Figure 7A). Each dilution response was tested in duplicate or triplicate. Twenty-three of these vaccinees’ sera and eight of the hybrid sera were tested twice and the mean values were calculated. The percentage of inhibition in the different groups of sera samples (median, IQR) gradually increased after the first dose-S2 (45.5, 26.5–69.5) and the second dose-S3 of the vaccine (96.8, 95.9–97.5), and then gradually decreased until 260d after the second dose—S4 (92, 73.6–96.3) and S5–S6 (62.5, 21.5–72.1). All sera tested after the third dose—S7 (97.3, 96.5–97.6)—up to 120 days later—(S8–S9) (96.6, 96.2–97.2)—including hybrid sera—S11 (95.7, 94.2–96.4)—obtained >90% percent inhibition.

Fifty-nine of the above 120 vaccinees’ sera and the hybrid sera were tested in serial dilutions and the ID_50_ inhibitory titer of each serum was calculated. Twenty three of these vaccinees’ sera and eight hybrid sera were tested twice, the two ID_50_ titer values were calculated and averaged. Note that the second sVNA test of hybrid sera was performed on the same ELISA plate simultaneously with the same number of sera from vaccinated individuals. As shown in Figure 7B, the ID_50_ titer values showed wide variation for sera samples S3, S7, S(8–9) and S11 in contrast to the values of the corresponding group samples tested at 1:20 dilution (Figure 7A). This observation confirmed the importance of producing serial dilutions from each sample instead of a single dilution. In details, ID_50_ values (median, IQR) gradually increased from S2 (21.6, 16.3–27.5) to S3 (341.3, 207.9–475.5) to S7 (638.5, 461.7–1004) and finally to S11 (1897.8, 1077.5–2486.5), whereas ID_50_ values of sera obtained at time points up to 260 days after the second dose-S4 (65.5, 42.4–108.9) and S5–S6 (28.6, 18.7–40.7) and up to 120 days after the third dose-S8–S9 (285.8, 165.3–567.6) decreased compared to S3 and S7, respectively. Also, it was observed that the ID_50_ titer values from the S11 sample group (14 hybrid sera) were higher compared to the S7 samples (Figure 7B), demonstrating that healthy vaccinated individuals were directly able to produce even greater anti-spike immunity for the original Wuhan strain. During this time period (June 2022–September 2023), Omicron strains were predominant in Greece (https://eody.gov.gr, accessed on 7 December 2023).

### 3.4. Comparison of nAbs Titers in Hybrid Sera with sVNA and cPass

In addition, eleven of the fourteen hybrid sera were also tested with cPass and inhibitory ID_50_ titer values calculated and compared with the corresponding averaged ID_50_ titer values of sVNA (nine of eleven samples were tested twice with sVNA). The analysis gave a weaker coefficient of determination for ID_50_ (R^2^ = 0.6230, *p* = 0.004), even excluding the one “outlier value” (R^2^ = 0.72, *p* < 0.01) (Figure 8A), when compared to the corresponding correlation of the ID_50_ values calculated from sera of vaccinated individuals (R^2^ = 0.9195, *p* < 0.0001) (Figure 6C). The inhibition curves of hybrid sera by sVNA and the corresponding inhibition curves generated from cPass are shown in Appendix A respectively, along with their ID50 values. Note that eight of these sera were performed on the same ELISA plate simultaneously with sera from vaccinees.

Interestingly, when hybrid sera were used, the sensitivity of sVNA was higher than the sensitivity of cPass (Appendix A), while it was lower when vaccinees’ sera were used (Appendix A). In order to compare the results from sVNA and cPass for hybrid sera more appropriately, we used the ratio of cPass-ID_50_: sVNA-ID_50_ titer values of sera from vaccinated individuals as control, against which the same ratio values of hybrid sera were compared. As a result, the ratio obtained by testing sera from vaccinated individuals with cPass and sVNA was 1.4 ± 0.09 (range 0.77 to 2.43) while the corresponding ratio of hybrid sera was 0.64 ± 0.06 (range of 0.3 to 1.0) (Figure 8B and Table 2). These results suggest that cPass-ID_50_ inhibition titer values were lower than sVNA values for hybrid sera (average fold change to sVNA is 3.17↓ to 1.0), while they were generally higher in sera from vaccinated individuals (average fold change to sVNA 1.05↓ to 2.18↑) (Table 2).

## 4. Discussion

Since the beginning of the COVID-19 pandemic, nAbs have been suggested to be the most promising biomarker for the evaluation of antiviral immune response and the protection achieved either by vaccination or natural infection. So far, the isolated RBD has been mainly used for the development of both commercial and “inhouse” sVNTs, based on the knowledge that the majority of nAbs target the RBD fragment of the S1 domain of SARS-CoV-2 spike protein [4,5,12].

In this study, we have developed an ELISA-based surrogate virus neutralization assay (sVNA) that can be completed within 3–4 h in a BSL2 laboratory. It utilizes the viral ectodomain trimeric spike S(6P)-StrepTag and the host cell receptor ACE2. Evaluation of the sVNA was performed with a single serum sample or serially diluted ones and the commercially available cPass neutralization test of GENSCRIPT. We proceeded to test a cohort of sera samples obtained at different time points after administration of the first, second and third doses of the Pfizer vaccine and from vaccinated patients with COVID-19 (hybrid sera), in order to evaluate their neutralizing effect. Finally, based on our experimental results from testing titrated sera with sVNA and cPass, we investigated whether the S(6P)-StrepTag molecule (sVNA) could measure more than RBD-RBM specific neutralizing antibodies, by comparing the ID_50_ titer values produced with these two inhibition assays.

Firstly, we studied the interaction between the trimeric S(6P), human neutralizing Abs, and immobilized h-ACE2 receptors under the view of minimizing the assay cut-off limit in parallel with higher sensitivity, reproducibility, and accuracy of the inhibition assay in development. Data from the optimization steps (S(6P)-StrepTag-ACE2 interaction, final sVNA protocol) showed that the FBS dilution agent performed better than n.f.-milk, resulting in higher assay sensitivity and a sigmoidal inhibition curve (Appendix A). The usage of n.f.-milk usually caused a divergence of the lower part of the titration curves from the norm, promoting a tendency for a higher cutoff limit of the inhibition curve (Appendix A). Taken together with a higher intra-assay coefficient of variability when n.f.-milk was used, the above data suggested that FBS was the most suitable reagent to be utilized in our sVNA.

Our results from testing vaccinated individuals’ sera with sVNA have shown a high reproducibility in either 1:20 or a serially diluted fashion, while the robustness of testing negative sera samples has been confirmed and a cut-off value of 20% is suggested, which would allow discrimination between negative and positive samples (Figure 5A–C). A representative serial dilution testing of sera samples that have been taken from five vaccinated individuals 21 days after the 1st, 2nd, and 3rd dose of the vaccine administration, showed desirable inhibitory curves (Figure 3A–C). The neutralizing activity of the samples was gradually increased with more vaccination doses, something which is in concordance with the results of other studies [23,57]. In addition, expected inhibitory curves were generated from all sera samples throughout the vaccination period of a person (S1–S10) that got infected with SARS-CoV-2 (S11–S12) after having received the third dose of the vaccine (Appendix A).

Secondly, we validated the inhibition assay by comparing the neutralization responses of the optimized sVNA-S(6P) to the responses within GENSCRIPT’s commercial cPass neutralization immunoassay, which detects nAbs that block the interaction between the isolated RBD and the ACE2. The reliability of cPass for the detection of nAbs response against SARS-CoV-2 has been extensively tested by others [21,22,25,50,51,52,54,58,59]. The findings of several studies support the use of the cPass assay to accurately detect the nAbs response against SARS-CoV-2, when the sera were tested either in a single dilution [25,51,52,54,58,60] or serially diluted [22,23,52,58].

When we tested sera from vaccinated individuals with different levels of nAbs, with both inhibition assays at the final dilution of 1:20 (Figure 6A), regression analysis showed a strong correlation between the two methods. Our results are in accordance with those of da Silva and colleagues [21], who recently developed an sVNT using the SARS-CoV-2 trimeric spike (S–2P) as a competitor, and used the cPass as well as a VNT to evaluate their assay. As titration results in better quantification of neutralizing antibody activity, and therefore in a more reliable degree of correlation [52], we proceeded to analyze the correlation between the neutralizing serum titers of the two assays. We found a high degree of correlation for vaccinated individuals’ sera between the two assays (Figure 6B,D), suggesting that sVNA is able to estimate neutralizing activity in titrated sera of vaccinated individuals and gives comparable results to cPass even in the presence of low nAbs levels. In another case, measurement of serially diluted sera from vaccinated individuals with a commercial kit that uses S(2P)-trimer as competitor demonstrated strong correlation with PRNT, suggesting that an sVNT can be used as a surrogate method for live virus neutralization [61,62]. To our knowledge, this is the first study to compare the inhibitory effect of vaccinees’ serum titers between two ELISA-based assays that used respective S-trimer or isolated RBD molecules.

Subsequently, we designed an experimental approach for using sVNA for longitudinal studies to evaluate nAbs in vaccinated individuals. A cohort of sera samples with different levels of nAbs (*n* = 127) were tested at a final dilution of 1:20, for their ability to inhibit the S(6P)-ACE2 interaction (Figure 7A). The inhibition rate gradually increased after the first (S2) and the second vaccine dose (S3), followed by a gradual decrease up to 260d after the administration of the second dose of the vaccine (S4, S5–6). All samples tested after the third dose (S7) and up to 200 days apart (S8–S9), including hybrid sera (S11), increased their neutralizing capacity (inhibition > 90%). These results are in line with numerous previous studies [47,57,63,64]. When samples from twelve of these vaccine recipients and all vaccinated patients with COVID–19 were tested in serial dilutions (Figure 7B), the inhibition ID_50_ values showed great variation for S3, S7, S(8–9), and S11 samples in contrast to the values of equivalent samples tested in 1:20 dilution (Figure 7A). Similar results have been reported by others using VNT assay [64,65]. The above comparison confirmed the importance of producing serial dilutions from each sample when dissecting the difference in % inhibition over time to assess the immune status of a vaccinated person or vaccinated patient with COVID-19. This approach, already underlined by others [52] as a result of using VNTs, arises, and is reported for the first time as a result of a titration analysis of sera performed with a surrogate VNT. 

The use of isolated RBD molecules in inhibition ELISAs to detect nAbs has the disadvantage of excluding those nAbs generated in vaccinated COVID-19 patients or vaccinated individuals, that target the entire S protein trimer of SARS-CoV-2 or the S1 and S2 domains of the S protein. The presence of such nAbs eventually contributes directly or indirectly to the successful inhibition of the trimeric S protein binding to ACE2 [4,5,12,13,14,15,66]. In line with this view, trimeric spike has been used instead of RBDs, but so far it remains unclear whether nAbs specific for epitopes outside the RBD-RBM region can be recognized. We found a high degree of correlation for ID_50_ titers from the analysis of vaccinated individuals’ sera (Figure 6C), while, in contrast, analysis of eleven hybrid sera showed a weak coefficient of determination for ID_50_ titers (Figure 8A). Further analysis showed that cPass ID_50_ titer values of vaccinees’ sera were higher than the corresponding ID_50_ values generated by sVNA (Appendix A). In contrast, titration of hybrid sera showed that cPass ID_50_ values were lower than the corresponding values produced by sVNA (Appendix A). As a result, when the eleven hybrid sera were tested, the ratio of cPass-ID_50_ to sVNA-ID_50_ inhibitory titers was significantly lower than the mean of the same ratio corresponding to the nineteen vaccinated individuals’ sera (Figure 8B and Table 2). Thus, an explanation for the above observation could be that our assay may identify and measure non-RBD-RBM nAbs that either exist in vaccinees’ sera and are produced at a higher rate in hybrid sera, or that SARS-CoV-2 infection induces the production of these nAbs with new specificities. We acknowledge the limitation that our findings indirectly showed the effect of this type of nAbs (non-RBD-RBM), through the detection of their inhibitory action using S trimer assay (sVNA), which could not be detected with RBD assay (cPass). Some research studies have reported that there was a higher ID_50_ titer of RBD-RBM nAbs in hybrid sera than in vaccinated individuals [67,68]. This fact alone could not reverse the ratio values of cPass-ID_50_ to sVNA-ID_50_, because both assays detect the nAbs with this specificity. To our knowledge, only the data by da Silva imply the possible existence of other than RBD-related nAbs that can block binding of the S trimer to ACE2 [21].

Ultimately, all types of nAbs that recognize something other than the RBD-RBM epitopes (with the exception of S2–specific nAbs) result in the change of the S trimer structure and conformation [4,8,69,70,71] or the reduction of the RBDs in the open state on the spike molecule [13,14,15]. Thus, by reducing the RBD accessibility or inducing the closed RBD state, they ultimately decrease the affinity of the spike for ACE2. Although some other antibodies have also been isolated that induce the open RBD state of Spike [9,10], tested vaccinees’ or hybrid sera always contain neutralizing properties. The existence of this type of antibody generated in vaccinees or vaccinated patients with COVID-19 (Figure 9), which could alter the strength of trimeric Spike-ACE2 interaction, might explain the above significantly higher (up to 2.18) or lower (up to 3.17) titer fold change of cPass to sVNA in vaccinees and vaccinated patients with COVID–19, respectively (Table 2). In an ELISA-based inhibition assay, reducing the number of accessible RBDs (isolated or as regions of the trimeric S molecule) leads to a higher sensitivity of the assay, and thus to a higher value of the inhibition titer ID_50_ (Figure 2). Although the majority of neutralizing antibodies have been shown to target the RBD “up” conformation [4,6,7] and their neutralizing mechanism is achieved by blocking the interaction between RBD-RBM and ACE2, it is possible that the final inhibitory activity of the pool of neutralizing antibodies tested using the S(6P)-StrepTag trimer in the sVNA may be disproportionately altered by the presence of antibodies with the above characteristics in the serum. To strengthen the above interpretation, further experiments can be performed, such as generating well-characterized specific mAbs from patients with COVID-19 as non-RBD-RBM nAbs and testing them with sVNA. But, that was outside the aim of this study. As already mentioned, several other research groups have produced and tested positive mAbs of this type, by live virus neutralization assay.

The use of sera from fully vaccinated patients with COVID-19 (hybrid immunity) instead of naïve patients and the comparison with vaccinees’ sera may have offered an advantage to our assay. The SARS-CoV-2 infection can enhance vaccination-induced existing immunity [64], producing a higher level of those nAbs that were initially generated by vaccine administration (Pfizer, Moderna, Cambridge, MA, USA and Johnson & Johnson, New Brunswick, NJ, USA) and, in addition, new nAbs outside the RBD-RBM region, probably due to the native conformation of the spike and the immune response to virus infection (vaccine administration: Monderna, Pfizer, Jansen, Oberriet, Switzerland, and Astrazeneca, Cambridge, UK) [47]. Early in the COVID-19 pandemic, it was reported that nAbs generated following SARS-CoV-2 infection had more heterogeneous spike epitope specificity than nAbs generated by mRNA vaccines [67]. Recent findings highlight the existence of qualitative differences in nAbs induced by infection or vaccination [72,73]. Furthermore, vaccination and hybrid immunity have been shown to produce different NTD and RBD non-RBM nAbs from each other, targeting distinct epitopes on these fragments of the SARS-CoV-2 S protein [74]. Similarly, Quandt et al. [57] have shown that hybrid immunity (Omicron BA.1) primarily expands a broad B-memory cell repertoire against conserved S protein and RBD epitopes, rather than inducing strictly Omicron BA.1–specific B-memory cells, thus inducing broadly neutralizing antibodies compared to vaccinated individuals. Interestingly, the results of another research group showed that more RBD non-RBM specific antibodies with neutralizing properties are produced exclusively in vaccinated patients with COVID-19 than in vaccinees alone [75]. The RBD non-RBM nAbs described by several studies could inhibit the interaction of spike with ACE2 directly [76,77,78] or indirectly [71,76,79,80,81], when utilized in antigen binding and cell infection assays. 

By testing and comparing the inhibitory titers in a hybrid serum with the two assays, we cannot detect and separately calculate the activity of non-RBD-RBM nAbs, because there are a variety of factors that determine the final result, including the type and concentration of the competitor (trimeric spike or RBD). We can only measure higher (or lower) activity of non-RBD-RBM neutralizing antibodies detected with sVNA alone, compared to immunized sera samples, when the RBD nAbs activity of the respective sera is tested simultaneously with the cPass-RBD assay. Indirectly, the sVNA evaluation as suitable to detect non-RBD-RBM nAbs, simultaneously confirmed the increasing production of this type of nAbs by the hybrid immunity as reported by others [74,75]. According to the above, we cannot exclude the possibility that testing sera of vaccinated individuals with sVNA may detect nAbs other than RBD-RBMs, as is the case with hybrid sera, but experimental results so far cannot definitely support this notion. 

## 5. Conclusions

The use of the ectodomain of SARS-CoV-2 spike as a stabilized recombinant soluble molecule with six proline substitutions (HexaPro) and hACE2 instead of isolated RBDs, and hACE2 in an ELISA based inhibition assay, may provide a simulation closer to virus-cell interaction in vivo.The new assay, developed in this study, is indirectly shown to be able to detect non-RBD-RBM nAbs that induce the RBD “close” conformation or affect the S(6P)-StrepTag trimeric conformation.The use of sVNA showed that a breakthrough SARS-CoV-2 infection is probably the reason for individuals who were also vaccinated (hybrid sera) to either develop higher titers of non-RBD-RBM nAbs (anti-NTD, anti-SD, anti-RBD non-RBM nAbs) compared to vaccinated ones, or induce the production of these nAbs with new specificities.

## Figures and Tables

**Figure 1 vaccines-12-00914-f001:**
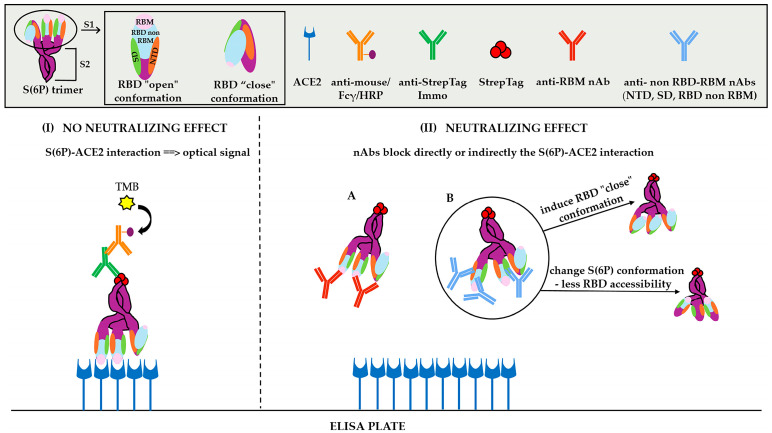
Graphical illustration of the developed surrogate Virus Neutralization Assay (sVNA). (**I**) Binding of SARS-CoV-2 trimeric spike S6(P)-StrepTag to hACE2, no nAbs; (**II**) (A) Direct blocking of S6(P)-StrepTag from binding to hACE2 by anti-RBM nAbs, (B) Indirect inhibition of S(6P)-ACE2 interaction by anti-RBD non-RBM nAbs, anti-NDT nAbs and anti-SD nAbs.

**Figure 2 vaccines-12-00914-f002:**
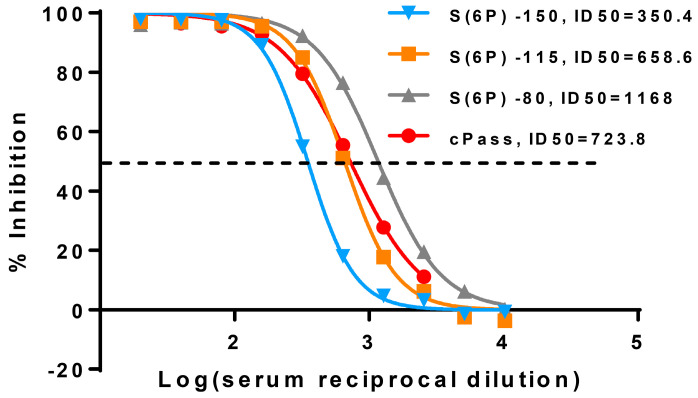
Titration curves of sVNA and cPass. Evaluation of the effect different concentrations of S(6P)-StrepTag have on the dose-dependent percent inhibition curves of a serum sample (S8). The red line represents the curve created from this sample using the commercial cPass test, that is similar to the curve (orange line) created by 115 ng/mL final concentration of S(6P) when using the sVNA. The experiment was repeated with three different sera in duplicates each time. A representative experiment is shown. The dots represent the mean values from these two replicates. The dotted line indicates the ID_50_* inhibition level. * The Inhibitory Dilution (ID) at which the serum was able to reduce the % inhibitory effects by 50% of the maximal S(6P)-ACE2 interaction, observed in the absence of serum, was designated as the sVNA-ID_50_ value.

**Figure 3 vaccines-12-00914-f003:**
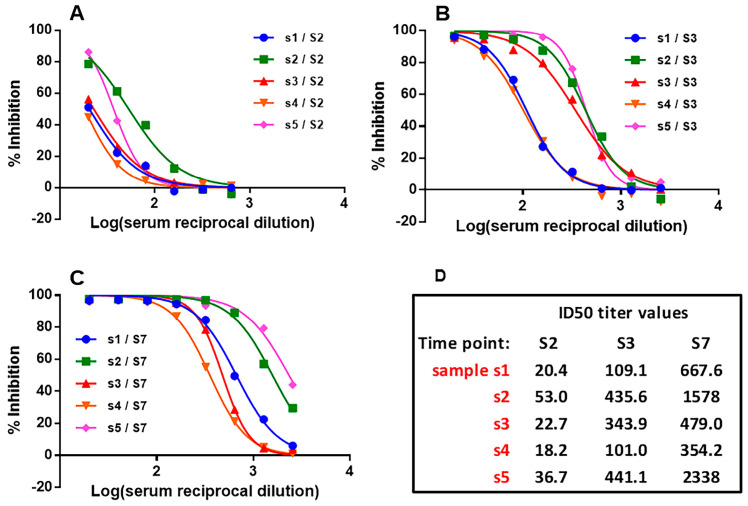
sVNA titration curves. Dose dependent inhibition of the interaction between immobilized ACE2 and soluble S(6P) in the presence of 5 representative sera (s): (**A**) S2 (21d after 1st dose of the vaccine), (**B**) S3 (21d after 2nd dose of the vaccine), (**C**) samples S7 (21d after 3rd dose of the vaccine). (**D**) ID_50_ values were determined as described in the method section, and are presented herein.

**Figure 4 vaccines-12-00914-f004:**
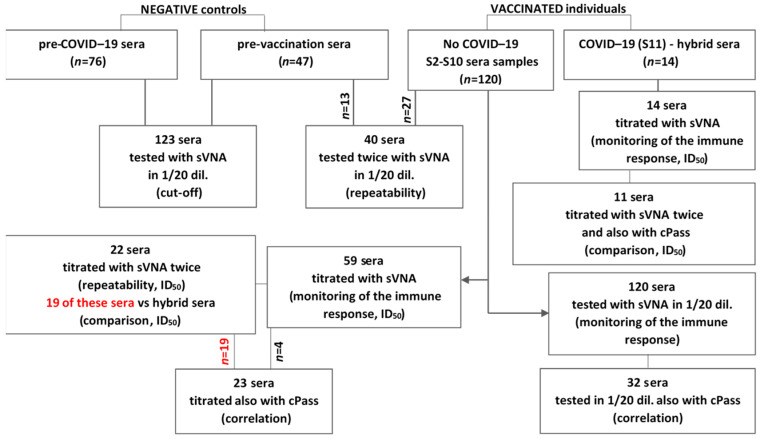
Recruitment, testing of sera and follow-up. In this study sera from non-vaccinated and vaccinated individuals (following administration of the first, second, and third dose of the BNT1−62b2 vaccine), as well as fully vaccinated patients infected with COVID−19 (hybrid sera) were used.

**Figure 5 vaccines-12-00914-f005:**
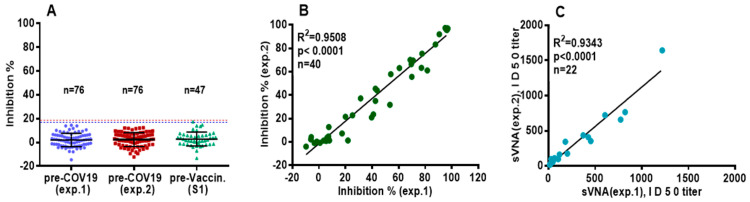
Testing sera with sVNAssay. (**A**) Sera of pre-COVID-19 (*n* = 76) at the final dilution of 1:20, were tested twice on two different experiments, and sera of pre-vaccinated individuals (S1, *n* = 47) at the final dilution of 1:20 were tested once. The horizontal lines indicate the Mean values ± SD. The dotted lines (3 × SD) represent the cutoff at ~20% Inhibition. (**B**) Sera from vaccinated (*n* = 27) and pre-vaccinated (*n* = 13) subjects were tested, at dilution 1:20, on two different experiments for their neutralizing ability and a correlation of their inhibition values was performed. (**C**) Twenty-two vaccinees’ sera were tested, in serial dilutions, on two different experiments and a correlation of their ID_50_ values was performed.

**Figure 6 vaccines-12-00914-f006:**
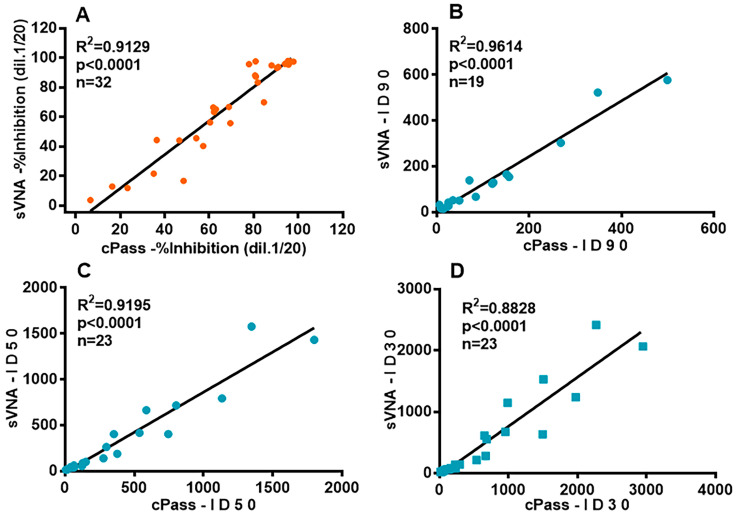
Correlation analysis for vaccinees’ sera with different levels of nAbs by sVNA and cPass assay. (**A**) Thirty-two sera from vaccinated individuals were tested in duplicate by sVNA and cPass, at a final serum dilution of 1:20. Mean values are shown. (**B**–**D**) The data presented are the ID_90_, ID_50_ and ID_30_ titer values, respectively, from twenty-three vaccinees’ serially diluted sera (with inhibition values ranging from 50% to 100% at 1:20 final dilution), tested with sVNA (the 19 sera titrated twice in two different experiments and the two ID_50–90–30_ titer values were averaged (see Figure 5C)) or cPass, with each dilution response tested in duplicate or triplicate. In panel 3B, four of the twenty three ID_90_ values could not be calculated, because of low nAbs titers.

**Figure 7 vaccines-12-00914-f007:**
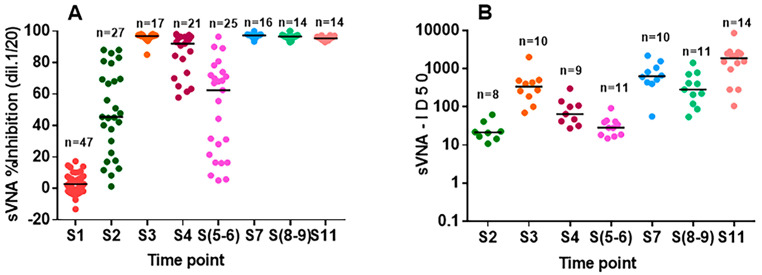
Testing sera samples derived from vaccinated persons with sVNA. (**A**) Time-course of neutralizing anti-S(6P) antibodies. Inhibition (%) of sera samples (S1–S11) at the final dilution of 1:20, collected before (S1), throughout the vaccination period (S2–S9) and after COVID-19 disease (S11). The horizontal lines indicate the median values. (**B**) Fifty-nine vaccinees’ sera collected from twelve individuals (all samples collected after the first vaccine dose (S2) induced >50% inhibition at the final dilution of 1:20) throughout the vaccination period, and fourteen patients’ sera were tested in serial dilutions and their ID_50_ values are presented. Indication of different samples refers to the status of each serum cohort (Section 2, Table 1).

**Figure 8 vaccines-12-00914-f008:**
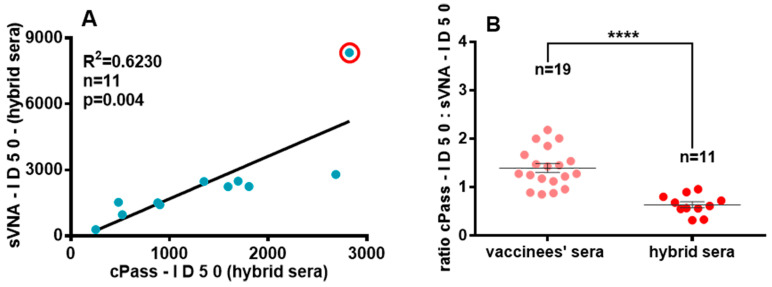
Testing vaccinated individuals and hybrid sera with sVNA and cPass. (**A**) Correlation of ID_50_ titer values of eleven hybrid sera. In the red circle, the “outlier value”. (**B**) Differences between the ratios of cPass-ID_50_: sVNA-ID_50_ titer values from fully vaccinated individuals and hybrid sera (**** *p* < 0.0001). The whole group of values are normally distributed; a two-sample *t*-test was used. Averaged mean values ± SEM are shown.

**Figure 9 vaccines-12-00914-f009:**
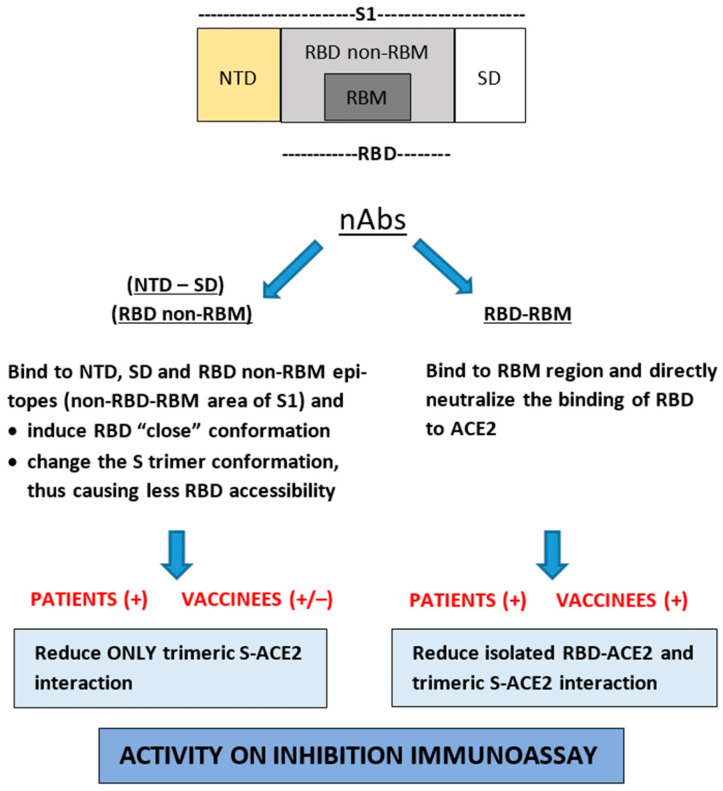
Schematic representation of nAbs targeting different S1 subdomains of SARS-CoV-2 Spike protein [RBD (receptor binding domain, consisting of RBM and non-RBM region), NTD (aminoterminal domain), SD (subdomain-intermediate region)] and their potential effect in the ectodomain S-trimer and isolated RBD interaction with ACE2.

**Table 1 vaccines-12-00914-t001:** Samples (S) of blood collection from vaccinated individuals (S11–S12 * hybrid sera).

Pre-Vaccine 0d	1st Dose 21d after	2nd Dose 21d	2nd Dose 60–65d	2nd Dose 190–200d	2nd Dose 240–260d	3rd Dose 21d	3rd Dose 60–65d	3rd Dose 90–120d	3rd Dose ~200d	COVID-19 15–25d
S1	S2	S3	S4	S5	S6	S7	S8	S9	S10	* S11

**Table 2 vaccines-12-00914-t002:** Comparison of ID_50_ values of vaccinated individuals’ sera with different levels of nAbs tested by sVNA (with >40% Inhibition at 1:20 dil.) and cPass and comparison of ID_50_ values of hybrid sera, tested by sVNA and cPass. Colored in blue are the inhibition values produced by sera run on the same ELISA plate. Arrows ↑ and ↓ indicate whether the value of Average Fold Change of cPass is higher or lower than sVNA ID_50_ titer, respectively.

Vaccine Name and Schedule	Serum * Sample (S)	cPass ID_50_ Titer	Average cPass-ID_50_/sVNA-ID_50_ (exp1–exp2)	*** Average Fold Change of cPass to sVNA ID_50_
BNT162b2 × 3 (*n* = 19)	S2	47.8	1.18 (1.01–1.34)	1.18↑
	S3	744.2	1.85 (1.99–1.71)	1.85↑
	S3	376.7	2.0 (1.86–2.14)	2.0 ↑
	S3	298.1	1.25 (0.87–1.64)	1.25↑
	S4	149	1.47 (1.42–1.53)	1.47↑
	S4	40.2	0.96 (1.05–0.87)	1.05↓
	S4	276.1	2.01 (1.70–2.33)	2.01↑
	S5	62.2	1.45 (1.22–1.68)	1.45↑
	S5	41.1	1.22 (0.99–1.45)	1.22↑
	S5	62.1	1.67 (1.23–2.11)	1.67↑
	S7	351.2	0.88 (0.99–0.77)	1.15↓
	S7	586	0.89 (0.81–0.97)	1.13↓
	S7	121.3	2.18 (1.92–2.43)	2.18↑
	S7	1344	0.85	1.17↓
	S8	1798	1.28 (1.09–1.47)	1.28↑
	S8	801.1	1.12 (1.21–1.03)	1.12↑
	S9	1132	1.43 (1.48–1.38)	1.43↑
	S9	125.7	1.54 (1.14–1.93)	1.54↑
	S10	536.2	1.28 (1.31–1.26)	1.28↑
		cPass-ID_50_/sVNA-ID_50_ = mean ± SEM→1.4 ± 0.09		
	** p1–S11	252.1	0.9 (0.79–1.0)	1.13↓
	p2–S11	2685	0.96 (1.0–0.93)	1.04↓
	p3–S11	1785	0.80	1.25↓
	p4–S11	1594	0.72 (0.66–0.78)	1.41↓
	p5–S11	1694	0.68	1.47↓
	p6–S11	901.2	0.61 (0.66–0.55)	1.66↓
	p7–S11	880.3	0.57 (0.53–0.60)	1.77↓
	p8–S11	1350	0.55 (0.52–0.57)	1.83↓
	p9–S11	521.4	0.56 (0.45–0.67)	1.85↓
	p10–S11	2823	0.33 (0.3–0.35)	3.06↓
	p11–S11	481.5	0.32 (0.34–0.3)	3.17↓
	cPass-ID_50_/sVNA-ID_5_ = mean ± SEM→0.64 ± 0.06			

* See Table 1. ** (p): fully vaccinated patients with COVID–19. *** Average fold change of cPass to sVNA ID_50_ is produced by the two-fold change values to sVNA, which resulted from the comparison of the two experiments sVNA ID_50_ with the cPass ID_50_ value (produced from one experiment).

## Data Availability

All relevant data are within the paper and its Appendix A.

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
