# Peer review of "Novel Competitive ELISA Utilizing Trimeric Spike Protein of SARS-CoV-2, Could Identify More Than RBD-RBM Specific Neutralizing Antibodies in Hybrid Sera"

_vaccines, 2024, doi:10.3390/vaccines12080914_

Round 1
Reviewer 1 Report
Comments and Suggestions for Authors
It is an interseting study, the authors developed competitive ELSIA assay to measure the titer of neutalizing antibodies, showing that the use of ectodomain of SARS-CoV–2 spike as a stabilized recombinant soluble 616 molecule with six proline substitutions (HexaPro) and hACE2 instead of isolated 617 RBDs and hACE2 in an ELISA based inhibition assay gives much better results and mimics the virus host interaction.
Major points:
1) The study lacks both in vitro study to support this findings. I understand that the tested samples from vaccinated patients, but still more in vitro and in vivo experiments should be done to confirm the findings.
2) All the vaccinated patients taking pfizer vaccine. What about other types of vaccines? did the author get the same findings?
3) demographic, clinical, vaccine data about the patients enrolled in the study in important for interpretation the results.
Comments on the Quality of English Language
Moderate language editing
Author Response
Please see the attachment.
Open Review
(x) I would not like to sign my review report
( ) I would like to sign my review report
Quality of English Language
( ) I am not qualified to assess the quality of English in this paper
( ) English very difficult to understand/incomprehensible
( ) Extensive editing of English language required
(x) Moderate editing of English language required
( ) Minor editing of English language required
( ) English language fine. No issues detected
|
Yes |
Can be improved |
Must be improved |
Not applicable |
|
|
Does the introduction provide sufficient background and include all relevant references? |
(x) |
( ) |
( ) |
( ) |
|
Is the research design appropriate? |
( ) |
(x) |
( ) |
( ) |
|
Are the methods adequately described? |
( ) |
(x) |
( ) |
( ) |
|
Are the results clearly presented? |
( ) |
(x) |
( ) |
( ) |
|
Are the conclusions supported by the results? |
( ) |
(x) |
( ) |
( ) |
Comments and Suggestions for Authors
It is an interesting study, the authors developed competitive ELSIA assay to measure the titer of neutralizing antibodies, showing that the use of ectodomain of SARS-CoV–2 spike as a stabilized recombinant soluble 616 molecule with six proline substitutions (HexaPro) and hACE2 instead of isolated 617 RBDs and hACE2 in an ELISA based inhibition assay gives much better results and mimics the virus host interaction.
Major points:
1) The study lacks both in vitro study to support this findings. I understand that the tested samples from vaccinated patients, but still more in vitro and in vivo experiments should be done to confirm the findings.
2) All the vaccinated patients taking pfizer vaccine. What about other types of vaccines? did the author get the same findings?
3) demographic, clinical, vaccine data about the patients enrolled in the study in important for interpretation the results.
Comments on the Quality of English Language
Moderate language editing
Comment-1: The study lacks both in vitro study to support these findings. I understand that the tested samples from vaccinated patients, but still more in vitro and in vivo experiments should be done to confirm the findings. Thanks for suggesting the utility for more experiments.
Answer-1: Thank you for pointing out the need for more experiments.
Competitive ELISA is a low variability type of ELISA that tests for the presence of an antigen-specific antibody in the test serum. We developed a novel inhibitory ELISA (sVNA) based on the use of the trimeric S(6P) molecule, which was positively evaluated with a high correlation rate when sera samples from vaccinated individuals were tested and compared to cPass, which uses RBD as a competitor molecule. However, the corresponding comparison in hybrid sera showed a significantly lower correlation percentage. More specifically, comparison of the cPass-ID50/sVNA-ID50 ratio between the two serum groups showed that in hybrid sera there is an increased ID50 titer from sVNA compared to cPass, whereas in contrast the sVNA-ID50 titer is decreased compared to cPass for most vaccinees’ sera. The only hypothesis that could explain the above result is the reduction of available RBD molecules in immunoassay, which would lead to an increase in its sensitivity (Fig.2). This may occur by the presence and effect of these specific neutralising antibodies in serum, which, as reported in bibliography, can reduce the number of open-form RBD and alter the structure of the trimeric spike, that either could change the RBD availability.
We agree that further experiments can be performed to strengthen the above interpretation, such as generating well-characterized specific mAbs from patients with COVID-19 as non-RBD-RBM nAbs and testing them with sVNA. But that was outside the aims of this study. Several other research groups have produced and tested positive mAbs of this type, by live virus neutralization assay [4,5,8,9,10,15]. However, we believe that our results, together with the bibliography, largely demonstrate the ability of sVNA to detect the activity of such non-RBD-RBM nAbs as well.
We acknowledge that our findings are indirectly explain the detection and effect of non RBD-RBM specific neutralizing antibodies and we revised our manuscript accordingly to emphasize this limitation in Discussion (9th paragraph, lines 642-647).
Comment-2: All the vaccinated patients taking pfizer vaccine. What about other types of vaccines? did the author get the same findings?
Answer-2: Thank you for pointing this out. We indeed tested the sera samples of individuals vaccinated with 3 doses of Pfizer vaccine (m-RNA vaccine) and subsequently infected with the virus (hybrid sera). We did not test sera from individuals who were vaccinated with any of the other approved vaccines and subsequently infected with COVID-19, nor are there any comparative studies in the bibliography –as far as we know- reporting results of testing their sera for the quality and characteristics of nAbs. However, what is highlighted by many publications are the different characteristics that have been found, between nAbs induced by vaccination or vaccination and SARS-CoV-2 infection. These studies have shown similar results regardless of the vaccine used and are already included in references ([63]: Pfizer-moderna-johnson & Johnson, [47]: monderna-pfizer-jansen-astrazeneca, [71]: Pfizer-moderna). It seems that administration of different vaccine type is independent in terms of the nAbs generated when compared to nAbs generated in hybrid sera. We revised our manuscript accordingly to point this out in Discussion lines 648-655.
Comment-3: demographic, clinical, vaccine data about the patients enrolled in the study is important for interpretation the results.
Answer-3: We thank the reviewer for helping us improve our manuscript.
Vaccinees’ sera samples (120) collected at 9 time points from 29healthy individuals vaccinated with the BNT162b2 vaccine (Pfizer) and 14 samples collected from 3–dose Pfizer vaccinees who subsequently contracted COVID–19, were used in this study to test nAbs with sVNA. The 29 vaccinated were healthy individuals of Greek origin aged 23—63. Twenty-one of them were female and 8of them male. Fourteen of them were classified as young adults (23—40) while 15were middle-aged (41—63). Nine of the 14 patients with COVID–19 were female, 5 of them were male. All vaccinees infected with SARS-CoV–2 became mildly or moderately ill, with no one being hospitalized. We have added the above text-information to the Material and Methods, (2.2.1., 1st paragraph, lines 642-649).

Reviewer 2 Report
Comments and Suggestions for Authors
In this paper, Eliadis et al. built novel competitive ELISA utilizing trimeric Spike protein of SARS-CoV-2 for identify neutralizing antibodies. It is important for other than RBD-RBM specific neutralizing antibodies in hybrid sera.
1. Is the His tag used for protein purification removed after protein purification? What is the success rate of resection? What is the purity of the final obtained protein? Is HPLC used for detection?
2. The formation of S protein trimer is a key part of this study. Has the formation of trimer been detected? What is the proportion of trimeric proteins in the correct form in the formed trimer? How to conduct testing? What are the test results?
3. Since detecting non RBD-RBM specific neutralizing antibiotics is an important objective of this study, how is the direct binding of non RBD-RBM specific neutralizing antibiotics to trimeric S protein to alter trimeric conformation validated through experiments? Is there any other experiment besides comparing with cPass to verify this result?
4. What is the storage temperature of the reagents used in this method? How stable is it? Especially the trimer of S protein.
5. There is a punctuation error in this manuscript that needs to be corrected.
Author Response
Please see the attachment.
REVIEWER-2
|
Yes |
Can be improved |
Must be improved |
Not applicable |
|
|
Does the introduction provide sufficient background and include all relevant references? |
( ) |
(x) |
( ) |
( ) |
|
Is the research design appropriate? |
( ) |
( ) |
(x) |
( ) |
|
Are the methods adequately described? |
( ) |
(x) |
( ) |
( ) |
|
Are the results clearly presented? |
( ) |
(x) |
( ) |
( ) |
|
Are the conclusions supported by the results? |
( ) |
( ) |
(x) |
( ) |
Comments and Suggestions for Authors
In this paper, Eliadis et al. built novel competitive ELISA utilizing trimeric Spike protein of SARS-CoV-2 for identify neutralizing antibodies. It is important for other than RBD-RBM specific neutralizing antibodies in hybrid sera.
Comment-1: Is the His tag used for protein purification removed after protein purification? What is the success rate of resection? What is the purity of the final obtained protein? Is HPLC used for detection?
Answer-1: We thank the reviewer for this comment. The His tag was used only for ACE2 purification. The S(6P) protein was affinity purified via its strep tag and a streptactin column. The His tag, present before the Strep tag (see image below), was further used to detect the S-trimer in the sVNA assay and thus could not be removed. The His tag or any other C-terminal tag has not been found to affect the conformation of the protein, and the proteins used for structure determination contained the tags. In ELISA experiments the recombinant proteins were used after the purification, without removing the tag.
The eluted S(6P) protein was >98% pure based on the Coomassie blue-stained gel after the affinity purification (Figure S1B).
Comment-2: The formation of S protein trimer is a key part of this study. Has the formation of trimer been detected? What is the proportion of trimeric proteins in the correct form in the formed trimer? How to conduct testing? What are the test results?
Answer-2: We thank the reviewer for this comment and we revised our manuscript accordingly by adding information.
The Hexa Pro Spike variant was designed and well characterized by Hsieh et al. (19), that it forms trimers. As all other class I fusion proteins, Spike is synthesized and secreted from cells as a trimer.
A codon-optimized nucleotide fragment encoding a stabilized version of the SARS-CoV-2 S ectodomain (amino acids 1 to 1208) followed by a foldon trimerization motif and tags (8×HisTag, StrepTag, and AviTag) was synthesized and cloned into pcDNA3.1/Zeo(+) expression vector (Thermo Fisher Scientific) by L. Grzelak et al. [44]. After trimeric S glycoproteins were produced by transient cotransfection of exponentially growing Freestyle 293-F cells and recombinant S proteins were purified, the investigators evaluated protein purity and formation by SDS–polyacrylamide gel electrophoresis in reducing and non-reducing conditions using NuPAGE 3 to 8% tris-acetate gels [44].
We used the above plasmid construct to produce recombinant S(6P) protein. S(6P) trimer formation was detected in the final eluted product material of the anti-strep column purification. We evaluated the eluted protein in an analytical run by gel filtration analysis using the Superose 6 Increase 10/300 GL column (GE Healthcare, Sweden) on an Akta FPLC purifier. A graph of the eluted protein is now added to Suppl. Materials, with the main peak corresponding to a molecular mass between 669 and 440 KDa. In result protein is expressed mainly as a trimer since the theoretical molecular mass of the monomer is 144 KDa and that the protein expressed is glycosylated. Furthermore, we performed S(6P) eluted protein of the anti-strep column purification on precast gradient gel 4-20%. An image of the gel that shows the purity and homogeneity of the used material is also added to Suppl. Material (Figure S1A-B). We have introduced more details in our manuscript accordingly: Introduction (lines 112-114) and Material and Methods (2.2.) (lines 179-182 and 192-194) and Suppl. Material (S2.3.).
We did not perform any experiments to confirm the correct conformation of the S(6P) trimer. This was done for this molecule and published by Hsieh et.al [19]. In this study, the inhibitory effect of nAbs in the sera of vaccinated individuals and in hybrid sera was comparatively studied. The results were derived from the effect of these sera on the same antigenic material of S(6P), so any differences observed are considered real and reliable.
Figure S1. (A) Gel filtration analysis of the expressed protein. The markers indicate the elution volumes of protein markers with known molecular weight (thyroglobulin: 669 KDa, ferritin: 440 KDa and aldolase: 158 KDa). Absorbance was detected at 280 nm and was expressed in mAU. The main peak of the eluted protein corresponds to the molecular mass of a trimeric state. (B) S(6P) purified protein performed on precast gradient gel 4-20%.
Comment-3: Since detecting non RBD-RBM specific neutralizing antibodies is an important objective of this study, how is the direct binding of non RBD-RBM specific neutralizing antibodies to trimeric S protein to alter trimeric conformation validated through experiments? Is there any other experiment besides comparing with cPass to verify this result?
Answer-3: We thank the reviewer for this comment. Other studies have shown the direct binding of non RBD-RBM nAbs to either the trimeric S protein of the virus or to recombinant protein molecules [15,71]. In our study, the effect of this type of nAbs was indirectly shown through the detection of their inhibitory action using S trimer assay (sVNA), which could not be detected with RBD assay (cPass). Using hybrid sera the sVNA gave higher sensitivity compared to cPass. In contrast, when vaccinees' sera were used, a similar or lower sensitivity was achieved with sVNA compared to cPass. This increase in sVNA (S-trimer) sensitivity can only be produced when the availability of an appropriate RBD form in the trimeric S(6P) is reduced. This can occur after changing the conformation of the S trimer (Fig.2) by non RBD-RBM nAbs. We acknowledge that our findings are indirectly explain the detection and effect of non RBD-RBM specific neutralizing antibodies and we revised our manuscript accordingly to emphasize this limitation (lines 614-617, 641-646, 686).
Comment-4: What is the storage temperature of the reagents used in this method? How stable is it? Especially the trimer of S protein.
Answer-4: Thank you for pointing this out. We agree with this comment. Hsieh et al. (19) looked well at the stability of S(6P), and it is more stable than the WT and than the 2P which was used for vaccines.
The storage temperature of the reagents used in this method is detailed in the Material and Methods (2.1.-2.2.) and in the Suppl. Material (S2.3.). Sera samples were divided into aliquots and stored at –80°C to avoid freeze thaw cycles. Recombinant proteins (ACE2 and S-trimer) were divided into aliquots and stored at –80 °C. Spike protein was also appropriately diluted in FBS at a final volume of 20μl and stored at–80 °C, as this approach was giving it strong antigenic stability. We tested the ability of these S(6P)-FBS samples to interact with immobilized ACE2 after 2 years of storage at –80 °C, compared to S(6P)-PBS samples stored for 2 years at –80 °C and S(6P) newly expressed and purified protein. Results from ELISA experiments showed exactly the same antigenicity characteristics between freshly purified samples and S(6P)-FBS samples, in contrast to S(6P)-FBS and S(6P)-PBS samples while less antigenicity was achieved with S(6P)-PBS samples.
The ACE2 protein was the most sensitive molecule, and after six to eight months its binding sensitivity to the S(6P) trimer decreased significantly. Six expression experiments for ACE2 were performed to complete the experiments and the product of two of them was rejected as inappropriate.
We have added the above information about the storage and the stability of S(6P) to the Supl. Material (S2.3.) (2nd paragraph).
Comment-5: There is a punctuation error in this manuscript that needs to be corrected.
Answer-5: We thank the reviewer for helping us improve our manuscript. We have corrected and revised any inaccuracies we found in the text. We also used dashes according to journals’ instructions: letter by letter (-) (normal), number-number (—) (Em-dash) and letter-number (–) (En-dash).

Round 2
Reviewer 1 Report
Comments and Suggestions for Authors
no further comments
Comments on the Quality of English Languagefine
Author Response
The authors would like to thank reviewer-1 for accepting our revised manuscript without any further comments.
Reviewer 2 Report
Comments and Suggestions for Authors
None
Author Response
The authors would like to thank reviewer-2 for accepting our revised manuscript without making any further comments.